# mRNA Capture Sequencing and RT-qPCR for the Detection of Pathognomonic, Novel, and Secondary Fusion Transcripts in FFPE Tissue: A Sarcoma Showcase

**DOI:** 10.3390/ijms231911007

**Published:** 2022-09-20

**Authors:** Anneleen Decock, David Creytens, Steve Lefever, Joni Van der Meulen, Jasper Anckaert, Ariane De Ganck, Jill Deleu, Bram De Wilde, Carolina Fierro, Scott Kuersten, Manuel Luypaert, Isabelle Rottiers, Gary P. Schroth, Sandra Steyaert, Katrien Vanderheyden, Eveline Vanden Eynde, Kimberly Verniers, Joke Verreth, Jo Van Dorpe, Jo Vandesompele

**Affiliations:** 1OncoRNALab, Center for Medical Genetics, Department of Biomolecular Medicine, Ghent University, 9000 Ghent, Belgium; 2Cancer Research Institute Ghent (CRIG), 9000 Ghent, Belgium; 3Department of Pathology, Ghent University Hospital, 9000 Ghent, Belgium; 4Department of Diagnostic Sciences, Ghent University, 9000 Ghent, Belgium; 5Molecular Diagnostics Ghent University Hospital (MDG), Ghent University Hospital, 9000 Ghent, Belgium; 6Biogazelle, 9052 Zwijnaarde, Belgium; 7Department of Internal Medicine and Pediatrics, Ghent University Hospital, 9000 Ghent, Belgium; 8Illumina, San Diego, CA 92122, USA

**Keywords:** mRNA capture sequencing, RT-qPCR, formalin-fixed paraffin-embedded (FFPE) tissue, fusion gene, fusion transcript, sarcoma, alveolar rhabdomyosarcoma, undifferentiated round cell sarcoma

## Abstract

We assess the performance of mRNA capture sequencing to identify fusion transcripts in FFPE tissue of different sarcoma types, followed by RT-qPCR confirmation. To validate our workflow, six positive control tumors with a specific chromosomal rearrangement were analyzed using the TruSight RNA Pan-Cancer Panel. Fusion transcript calling by FusionCatcher confirmed these aberrations and enabled the identification of both fusion gene partners and breakpoints. Next, whole-transcriptome TruSeq RNA Exome sequencing was applied to 17 fusion gene-negative alveolar rhabdomyosarcoma (ARMS) or undifferentiated round cell sarcoma (URCS) tumors, for whom fluorescence in situ hybridization (FISH) did not identify the classical pathognomonic rearrangements. For six patients, a pathognomonic fusion transcript was readily detected, i.e., *PAX3*-*FOXO1* in two ARMS patients, and *EWSR1*-*FLI1*, *EWSR1*-*ERG*, or *EWSR1*-*NFATC2* in four URCS patients. For the 11 remaining patients, 11 newly identified fusion transcripts were confirmed by RT-qPCR, including *COPS3*-*TOM1L2*, *NCOA1*-*DTNB*, *WWTR1*-*LINC01986*, *PLAA*-*MOB3B*, *AP1B1*-*CHEK2,* and *BRD4*-*LEUTX* fusion transcripts in ARMS patients. Additionally, recurrently detected secondary fusion transcripts in patients diagnosed with *EWSR1*-*NFATC2*-positive sarcoma were confirmed (*COPS4*-*TBC1D9*, *PICALM*-*SYTL2*, *SMG6*-*VPS53*, and *UBE2F*-*ALS2*). In conclusion, this study shows that mRNA capture sequencing enhances the detection rate of pathognomonic fusions and enables the identification of novel and secondary fusion transcripts in sarcomas.

## 1. Introduction

Fusion genes are chimeric genes arising from a chromosomal translocation that fuses parts of two genes. Such genes have been shown to drive tumorigenesis in hematological disorders and solid tumors, such as bone and soft tissue sarcomas, and are prototypical examples of disease-defining genomic aberrations. As such, fusion gene detection has become an integral part of routine clinical practice for these diseases [1,2]. Over the past decades, detection methodologies have evolved from guided fusion gene identification, using fluorescence in situ hybridization (FISH), over targeted PCR-based approaches or high-throughput array-based technologies, to unbiased fusion gene detection using massively parallel sequencing [2]. This technical evolution dramatically reshaped the fusion gene landscape due to the rapid identification of previously unknown fusion genes. Today, over 33,000 unique fusion genes have been described in cancer, targeting approximately 14,000 unique genes [3].

Different types of massively parallel sequencing technologies have been applied for fusion gene identification, using both DNA- and RNA-based approaches [4]. Whole-genome sequencing (WGS) enables precise determination of the genomic location of the fusion event, but it requires excessively deep sequencing and therefore is less suited for implementation in clinical diagnostics. The required sequencing depth can be drastically diminished by switching to RNA sequencing since only 1–2% of the genome is expressed as a protein-coding gene, resulting in reduced complexity of RNA sequencing libraries compared to WGS libraries. In addition to whole-genome and whole-transcriptome sequencing, target-enriched sequencing methodologies have been applied to improve fusion detection sensitivity, including hybridization capture sequencing and multiplex PCR-based methods [4,5,6,7,8,9,10,11,12,13,14]. Importantly, compared to DNA-based methods, RNA sequencing also provides insights into fusion transcripts generated by transcriptional readthrough of adjacent genes and can additionally be exploited for expression profiling, copy number variation, and sequence variant analyses.

In parallel to the advent of various RNA sequencing-based methodologies for fusion transcript identification, a plethora of specific bioinformatics software packages for fusion transcript calling was developed [15,16,17]. Most of these packages call fusion transcripts based on bridging read pairs that map to opposite sides of the fusion junction and/or split reads that directly overlap the fusion junction. Although the prediction accuracy of these packages varies considerably, in general, their use inevitably results in the detection of false-positive fusion transcripts [17]. Therefore, novel fusion transcripts identified by deep sequencing invariably require orthogonal method validation.

In this study, we aim to establish an optimized workflow for fusion transcript identification using RNA-based hybridization capture sequencing in FFPE tissue, the routinely used and most common resource of archived biomaterial in pathology departments. For this purpose, two Illumina mRNA capture sequencing methods, i.e., TruSight RNA Pan-Cancer Panel and TruSeq RNA Exome sequencing, are applied to FFPE tissue samples of sarcoma patients to identify both pathognomonic and novel fusions. Additionally, we provide an RT-qPCR-based workflow to validate the identified fusion transcripts.

## 2. Results

### 2.1. Orthogonal Validation of Known Fusions in Sarcoma Using mRNA Capture Sequencing

To validate our workflow for the identification of fusion transcripts using mRNA capture sequencing, we used TruSight RNA Pan-Cancer Panel profiling to analyze a first cohort (cohort I) of formalin-fixed paraffin-embedded (FFPE) biomaterials of six cancer patients for whom the diagnostic workup demonstrated the presence of a specific chromosomal rearrangement in the tumor (Figure 1 and Table 1). For all patients, RNA from both tumor and adjacent normal FFPE tissue was isolated (with DV200 values ranging from 14.5% to 56% (median of 45.5%); Appendix A), prepped, and sequenced using this panel, followed by fusion transcript analysis. All library preparations were successful, apart from the library preparation of the normal FFPE tissue of patient P16, for which no enrichment of target regions was observed in the Bioanalyzer smear analysis. This sample was excluded from further analysis. Detailed mapping statistics and identified fusion transcripts in each of the samples are given in Appendix A, respectively. As demonstrated in Table 1, mRNA capture sequencing confirmed all known chromosomal rearrangements in the tumor samples, with 3.52 to 30.69 (median 8.97) fusion supporting reads per million uniquely mapped reads. Furthermore, it enabled the identification of both fusion gene partners and defined fusion breakpoints (Table 1), which is impossible using fluorescence in situ hybridization (FISH) with break-apart rearrangement probes. None of the known aberrations were present in the matching normal tissue samples. These data validate the mRNA capture sequencing analysis workflow for the identification of fusion transcripts.

### 2.2. Unbiased mRNA Capture Sequencing Reveals Pathognomonic Fusion Transcripts in Clinicopathological Enigmatic Sarcomas

Next, RNA Exome sequencing was applied to a second cohort (cohort II) of FFPE tumor RNA of 17 patients diagnosed with alveolar rhabdomyosarcoma (ARMS) or undifferentiated round cell sarcoma (URCS), that were designated fusion gene-negative by FISH analysis (Figure 1). This means that the presence of the pathognomonic fusion gene was not demonstrated using the standard diagnostic workup. To gain insights into these clinical enigmas, FFPE tumor RNA was isolated (with DV200 values ranging from 10.20% to 75.50% (median of 28.20%)) and profiled using the validated pipeline (Appendix A). Strikingly, our mRNA capture sequencing analysis workflow identified a pathognomonic fusion transcript in 6/17 (35.29%) patients, detected with a read evidence level ranging from 0.36 to 1.73 (median 1.40) fusion supporting reads per million uniquely mapped reads (Table 1 and Appendix A). This remarkable finding prompted us to further inspect the clinicopathological characteristics and FISH results of these patients (Appendix A). For the patients with ARMS (P18 and P25), we detected a *PAX3-FOXO1* fusion, whereas *FOXO1* break-apart FISH demonstrated 0% positive tumor cells, and patients were diagnosed based on clinicopathological and morphological findings only. For the URCS patients (P26–P29), we detected an *EWSR1-ERG*, *EWSR1-NFATC2*, or *EWSR1-FLI1* fusion, but the *EWSR1*, *FUS*, *CIC*, and *BCOR* break-apart FISH results were negative. This means that the percentage of positive cells is below the cutoff of 15% (10–12% for *EWSR1*) or that 0% positive tumor cells were detected (for *FUS*, *CIC*, and *BCOR*). Based on clinicopathological, morphological, and immunohistochemical findings only, P26 and P29 were diagnosed as Ewing sarcoma, and P27 and P28 as undifferentiated round cell sarcoma (small cell osteosarcoma, Appendix A).

### 2.3. RT-qPCR Validates Pathognomonic Fusion Transcripts Detected by mRNA Capture Sequencing in FISH-Negative Sarcomas

To confirm the mRNA capture sequencing results, an RT-qPCR-based validation workflow was set up. First, fusion transcript assays for the pathognomonic fusions detected in cohort II (Table 1) were manually designed using the Shiny app DNA Melting Thermodynamic Model, and in vitro validated using six serial dilutions (from 1,000,000 molecules/µL to 10 molecules/µL) of 60-mer synthetic oligonucleotides. All assays showed to have excellent PCR amplification efficiency (between 98.3% and 103.1%). Secondly, the optimal FFPE cDNA input amount for qPCR was determined using 3 reference assays (*Alu-Sq*, *Alu-Sx*, and *Alu-J*) and a dilution series of normal FFPE tissue RNA of patient P15, i.e., the cohort I patient with the highest normal FFPE tissue RNA yield, enclosing six 10-fold dilution points ranging from 2.5 ng cDNA to 0.000025 ng cDNA per reaction. Based on the relatively high Cq values of the first dilution point (18.43 to 20.13), 4 ng cDNA was put forward as input for RT-qPCR. As such, the pathognomonic fusion transcripts detected in cohort II could be validated using RT-qPCR, with Cq values ranging from 27.01 to 34.69 (Figure 1 and Appendix A).

### 2.4. Novel Fusion Transcripts Are Identified and Validated in Clinicopathological Enigmatic Sarcomas

Apart from the pathognomonic fusion transcripts, 7 to 80 (median of 50) additional fusion transcripts were identified in each of the patients of cohort II (Appendix A). For patients with no pathognomonic fusion transcript detected and for which the molecular basis of tumorigenesis thus remains unclear, this list of additional fusion transcripts was filtered and prioritized to exclude falsely identified fusion transcripts and to select candidates for RT-qPCR validation (see Section 4). Initially, 20 RT-qPCR assays for 20 fusion transcripts, designed using primerXL, were in vitro validated on synthetic templates and subsequently tested on patient material (Appendix A). In total, the presence of nine fusion transcripts was validated (highlighted in light gray in Table 2), with Cq values ranging from 30.43 to 36.28 (Appendix A). For the 11 other fusion transcripts, RT-qPCR resulted in amplification of at least one of the control samples (3/20) or no amplification at all (8/20). For the eight fusion transcripts for which the primerXL assays did not show amplification, new RT-qPCR assays were designed using the Shiny app DNA Melting Thermodynamic Model (Appendix A) and analogously tested on synthetic templates and tumor samples. One of these assays could not be in vitro validated on synthetic templates, and of the remaining seven assays, two were validated (highlighted in dark gray in Table 2) with Cq values ranging from 33.50 to 35.49 (Appendix A), bringing the total to 11/20 (55%; Figure 1). Remarkably, for all RT-qPCR confirmed fusion transcripts, the fusion junction coincides with a known exon border for both the 5′ and 3′ fusion partner. All these fusion transcripts, except *IGK@*-*BAGE2*, are intrachromosomal fusion transcripts (Table 2).

### 2.5. RT-qPCR Confirms Recurrently Detected Secondary Fusion Transcripts in Sarcomas with an EWSR1-NFATC2 Fusion

Apart from the identification of novel fusion transcripts in enigmatic sarcomas, the obtained fusion transcript lists were also screened for recurrently detected fusions besides the pathognomonic fusions in ARMS and URCS patients (see Section 4). At the gene level, three recurrent fusions (*ATXN3*-*THAP11*, *AC245595.1*-*IGK@*, and *ELMO1*-*AOAH*) were detected in ARMS, and four (*COPS4*-*TBC1D9*, *PICALM*-*SYTL2*, *SMG6*-*VPS53*, and *UBE2F*-*ALS2*) in URCS (Appendix A), more specifically in patients with an *EWSR1*-*NFATC2* fusion. For RT-qPCR validation, only recurrently detected transcripts (transcript level) with fusion events at exon-exon borders were selected. For six fusion transcripts (Table 3) RT-qPCR assays were designed with the Shiny app DNA Melting Thermodynamic Model (Appendix A), in vitro validated on synthetic templates, and subsequently tested on patient material. While the secondary fusions in the ARMS patients could not be validated, for the *EWSR1*-*NFATC2*-positive patients, the presence of the four secondary transcripts was confirmed (highlighted in gray in Table 3), with Cq values ranging from 27.15 to 35.55 (Figure 1 and Appendix A). Three of them (*COPS4*-*TBC1D9*, *SMG6*-*VPS53*, and *UBE2F*-*ALS2*) could not be detected in the other *EWSR1*-rearranged patients of cohort II and are thus specifically expressed in sarcomas with an *EWSR1*-*NFATC2* fusion.

## 3. Discussion

We assessed the performance of two mRNA enrichment sequencing methods, i.e., TruSight RNA Pan-Cancer Panel and TruSeq RNA Exome sequencing, for fusion transcript identification in FFPE tumor tissue. Although both panels have previously been reported to detect pathognomonic fusions [8,18], they have not been extensively evaluated for unbiased fusion transcript identification. To robustly detect both known and novel fusion transcripts, we made use of the FusionCatcher software package [17,19,20,21] in combination with RT-qPCR validation.

First, the workflow was optimized using the TruSight RNA Pan-Cancer Panel, targeting 1385 cancer genes, in a cohort of sarcoma FFPE samples harboring a pathognomonic fusion gene, as determined by FISH. With a median detection sensitivity of 8.97 fusion supporting reads per million uniquely mapped reads, all fusions were confirmed. Note that for patients P12 and P16 multiple fusion transcripts (with different fusion partners) were called by FusionCatcher. For patient P12, two different chromosomal positions of the 5′ end of the fusion junction were identified, resulting in two different fusion transcript sequences for the *EWSR1*-*FLI1* fusion. This finding is in line with the reported simultaneous detection of multiple fusion transcripts in the same tumor caused by *EWSR1* splice variants in Ewing sarcoma [22]. Although the co-existence of different fusion genes and transcripts have also been reported in synovial sarcoma [23,24], the two fusion transcripts (*SS18*-*SSX2* and *SS18*-*SSX2B*) identified for patient P16 likely result from an alignment artifact due to sequence similarity between SSX2 and SSX2B (99.97% BLAST identity), which is supported by the prediction of an identical fusion transcript sequence by FusionCatcher (Appendix A). Importantly, in contrast to break-apart FISH, fusion gene detection by mRNA capture sequencing also enabled the identification of both fusion partners, which is essential for prognostication and diagnostic accuracy. For example, it was previously shown that *PAX3*-*FOXO1*-positive ARMS patients have worse overall survival rates compared to *PAX7*-*FOXO1*-positive ARMS patients [25].

Upon validation of our analysis workflow on sarcoma samples with a known pathognomonic fusion, a broader enrichment method, TruSeq RNA Exome targeting 21,000 mRNAs, was applied to the second cohort of 17 enigmatic ARMS and URCS patients with no pathognomonic fusion detected by FISH. Surprisingly, a pathognomonic fusion transcript was readily identified in six of these patients, with a median detection sensitivity of 1.40 fusion supporting reads per million uniquely mapped reads. This underscores the high sensitivity of mRNA capture sequencing to identify fusions in low-quality FFPE samples. For patients P26–P29, the discrepancies between the *EWSR1* break-apart FISH and sequencing results can be explained by borderline negative FISH results (below validated cutoff). The *FUS*, *CIC,* and *BCOR* break-apart FISH for these patients showed 0% positive tumor cells, which is in line with the identified fusion transcripts by mRNA capture sequencing. However, for patients P18 and P25, who showed to be *PAX3*-*FOXO1*-positive by mRNA capture sequencing, two independent *FOXO1* break-apart FISH analyses (performed at different pathology departments) demonstrated 0% positive tumor cells. Given that the fusion position in *FOXO1* predicted by mRNA capture sequencing for these patients is exactly the same as the one predicted for patients P13 and P14, two patients of cohort I that show positive *FOXO1* break-apart FISH, we assume that incompatibility between the fusion position and FISH probes is unlikely. Clearly, this points toward a higher detection sensitivity of mRNA capture sequencing. Of note, it might also be that the *PAX3*-*FOXO1* fusion is only present at the transcript level and that the corresponding chromosomal rearrangement is absent, as previously demonstrated in mesenchymal stem cells [26]. These cells transiently express *PAX3*-*FOXO1* as a result of a posttranscriptional process such as trans-splicing of precursor mRNAs, committing them to the myogenic lineage by transactivating the expression of the essential myogenesis factors *MYOD* and *MYOG*. When constantly expressed, *PAX3*-*FOXO1* interferes with the muscle differentiation process, which presumably contributes to tumorigenesis [26]. Further research should point out whether the expression of *PAX3*-*FOXO1* in the absence of a chromosomal rearrangement can truly cause ARMS. Interestingly, in two URCS patients, we detected an *EWSR1*-*NFATC2* fusion transcript. *EWSR1*/*FUS*-*NFATC2*-positive round cell sarcomas are relatively uncommon—worldwide, only 69 cases have been described so far—and patients demonstrate characteristic clinicopathological features delineating them from other Ewing and Ewing-like sarcoma patients, including older age at diagnosis and presence in the extremities [27,28,29,30]. The clinicopathological profiles of the two *EWSR1*-*NFATC2*-positive cases of cohort II (Appendix A) nicely confirm these previous findings. Clearly, as shown previously [18], mRNA capture sequencing may enhance the detection sensitivity of clinically important fusions, and therefore, implementing this technology in the diagnostic workup of sarcoma patients should be evaluated. In this context, the choice between RNA Pan-Cancer Panel profiling or RNA Exome sequencing is mainly determined by the budget that is available for profiling, with RNA Exome library preparation being more expensive than library preparation using the RNA Pan-Cancer Panel. Obviously, the main advantage of RNA Exome sequencing is that the analysis is not limited to a relatively small set of genes (i.e., 1385 cancer genes for the RNA Pan-Cancer Panel) but instead assesses 21,000 mRNAs. As such, from a clinical point of view, one should mainly answer the question of whether the additional data that are obtained using RNA Exome sequencing justifies the cost that comes along. Other profiling costs (lab and personnel costs) and turnaround times are comparable for both technologies.

Apart from pathognomonic fusion transcripts, also novel fusion transcripts can be identified using RNA sequencing. To reduce the risk of reporting results of false-positive novel fusion transcripts, results of multiple fusion callers are often combined to pinpoint a common set of transcripts. However, this still does not exclude the possibility of detecting false-positives and might even counteract the performance of highly accurate fusion callers by excluding true positives not identified by less accurate fusion callers. For example, the experimentally validated fusion *CSE1L*-*AL035685.1* in SK-BR-3 cells is only predicted by FusionCatcher and not by 22 other fusion callers [17]. Therefore, in this study, we opted to make use of a single fusion caller (i.e., FusionCatcher, which was previously shown to be fast and highly accurate [17,19,20,21]), combined with a literature-based prioritization strategy to select fusion transcripts for RT-qPCR confirmation in order to truly distinguish true and false-positive fusions. Nevertheless, it should be noted that the use of additional accurate fusion callers (such as STAR-Fusion and Arriba [17,31]) might also have led to the identification of additional pathognomonic fusions in the remaining patients of cohort II (i.e., patients that are false-negative by FusionCatcher), as well as to the identification of other potential clinically relevant novel fusions that are now excluded from the analysis. Using the pathognomonic fusion transcripts of the second cohort as test cases, an RT-qPCR validation pipeline was set up by designing primer pairs and first testing the assays on synthetic template dilution series. Note that throughout the study, we observed that only fusion transcripts reported as exon-exon fusion by FusionCatcher could be confirmed. When only considering the exon-exon fusion transcripts, the general validation rate of our study reaches 79.17% (19/24 assays) in contrast to the other validation attempts (55%, 11/20 assays). Therefore, we highly recommend prioritizing this type of fusions for RT-qPCR validation.

For patients in the second cohort without pathognomonic fusion transcript detected, novel fusion transcripts were identified by RNA Exome sequencing, with a median sequencing detection sensitivity of 0.39 fusion supporting reads per million uniquely mapped reads, and validated using the RT-qPCR workflow, including *COPS3*-*TOM1L2*, *NCOA1*-*DTNB*, *WWTR1*-*LINC01986*, *PLAA*-*MOB3B*, *AP1B1*-*CHEK2* and *BRD4*-*LEUTX* in ARMS patients. Interestingly, several gene partners of these fusion transcripts were previously reported to be important in ARMS or were described to be involved in rearrangements detected in other sarcoma types. For example, both fusion partners of the *COPS3*-*TOM1L2* fusion transcript have been described to be frequently amplified in osteosarcoma [32,33], and *TOM1L2* is involved in a recurrently detected rearrangement (*TOM1L2*-*BRAF*) in myxoinflammatory fibroblastic sarcomas [34]. *NCOA1* is a known fusion partner of *PAX3* (*PAX3*-*NCOA1*) in ARMS [35] and of *TEAD1* (*TEAD1*-*NCOA1*) in spindle cell rhabdomyosarcoma [36]. *WWTR1* is a downstream effector of the Hippo signal transduction pathway, a major player in different types of sarcoma, including ARMS [37,38,39,40], and in epithelioid hemangioendothelioma, *WWTR1*-*CAMTA1* has been identified as a disease-defining gene fusion [41,42,43,44]. Interestingly, in patient P19, the fusion transcript partner of *WWTR1* is a long intergenic non-protein-coding RNA (*LINC01986*). Also *MOB3B* functions in the Hippo signal transduction pathway, and alterations of this gene have been associated with different types of cancer [45]. In Ewing sarcoma, disruption of *AP1B1*, a neighboring gene of *EWSR1*, has been shown to be caused by bridged chromoplectic rearrangements also fusing *EWSR1* and *FLI1* [46]. Finally, also both gene partners of *BRD4*-*LEUTX* have been previously linked to sarcoma. An intrachromosomal *CIC*-*LEUTX* fusion was detected in angiosarcoma [47], and in ARMS, it was demonstrated that *PAX3*-*FOXO1* depends on *BRD4* to drive the expression of its target oncogenes and that *BRD4* inhibition leads to significant tumor suppression, ablating the transcription-driving function of the fusion gene [48]. Moreover, in a patient diagnosed with undifferentiated sarcoma with epithelioid morphology, also Barresi et al. recently identified a *BRD4*-*LEUTX* fusion transcript, though with different breakpoints for both fusion partners compared to the *BRD4*-*LEUTX* fusion transcript detected in patient P23 [49]. Additional studies are needed to define the oncogenic role and clinical value of the identified fusions, but the detection of these novel fusion transcripts could shed new light on the molecular events underlying tumorigenesis in clinically enigmatic ARMS tumors and open new horizons for precision therapy.

Finally, we also screened for recurrently detected fusion transcripts in the ARMS and URCS patients of cohort II. As such, in the *EWSR1*-*NFATC2*-positive patients, four secondary fusion transcripts (*COPS4*-*TBC1D9*, *PICALM*-*SYTL2*, *SMG6*-*VPS53*, and *UBE2F*-*ALS2*), all intrachromosomal fusions with fusion partners more than 200 kbp apart, were identified and confirmed by RT-qPCR, of which three (*COPS4*-*TBC1D9*, *SMG6*-*VPS53*, and *UBE2F*-*ALS2*) could not be detected in other EWSR1-rearranged tumors. This is in line with previous findings demonstrating that sarcomas with *EWSR1*-*NFATC2* fusions not only show characteristic clinicopathological features but also segregate from other types of URCS on a molecular level [29,50,51]. Since it was recently shown that, apart from the pathognomonic fusion, also secondary fusions may show oncogenic potential in *EWSR1*-rearranged sarcomas [52], it would be worthwhile to further investigate the functional role of *COPS4*-*TBC1D9*, *SMG6*-*VPS53*, and *UBE2F*-*ALS2* in *EWSR1*-*NFATC2*-positive patients.

Remarkably, some of the novel (secondary) RT-qPCR-confirmed fusion transcripts in the ARMS and *EWSR1*-*NFATC2*-positive URCS patients are out-of-frame fusions (*PITPNC1*-*CACNG4*, *PLAA*-*MOB3B*, *GON4L*-*SMG5*, *COPS4*-*TBC1D9*, and *UBE2F*-*ALS2*), target pseudogenes or long non-coding RNAs (*IGK@*-*BAGE2* and *WWTR1*-*LINC01986*) or involve the 5′ UTR of one of the fusion partners (*PICALM*-*SYTL2*). Although these transcripts do not form classical fusion proteins, these types of fusion transcripts may also exhibit biological functions, as previously exemplified in various types of cancer [53,54,55,56,57,58]. Further research on the functions of these transcripts in ARMS and URCS could discover alternative biological processes beyond the formation of oncogenic fusion proteins, contributing to sarcoma pathogenesis.

In conclusion, we developed a highly sensitive and reliable mRNA capture sequencing- and RT-qPCR-based workflow to identify and validate fusion transcripts in FFPE tissue samples. We showed that mRNA capture sequencing has a higher sensitivity than FISH by the detection of a known pathognomonic fusion in six FISH-negative sarcoma cases and confirmed the presence of four secondary fusion transcripts in *EWSR1-NFATC2*-positive patients. In addition, multiple novel fusion transcripts were identified in patients without pathognomonic fusion. In this study, sarcoma FFPE tissue was used to showcase the clinical applications of this methodological approach, but the proposed workflow can also be applied to other pathology FFPE archives.

## 4. Materials and Methods

### 4.1. Patient Biomaterial

Formalin-fixed paraffin-embedded (FFPE) biomaterials of two independent cohorts of 6 (cohort I) and 17 (cohort II) sarcoma patients were analyzed. Cohort I was profiled using the TruSight RNA Pan-Cancer Panel (Illumina, San Diego, CA, USA) and included FISH-positive patients diagnosed with alveolar rhabdomyosarcoma (ARMS, *n* = 2), Ewing sarcoma (*n* = 1), myxoid/round cell liposarcoma (*n* = 1) or synovial sarcoma (*n* = 2). Cohort II was analyzed using TruSeq RNA Exome sequencing (Illumina) and comprised FISH-negative patients diagnosed with ARMS (*n* = 9) or undifferentiated round cell sarcoma (URCS, *n* = 8). Hematoxylin and eosin staining of each of the tumors can be found in Appendix A (cohort I) or Appendix A (cohort II). Informed consent was obtained from each patient, and the study was approved by the ethical committee of the Ghent University Hospital (approval number 2004/094). All experiments were performed following relevant guidelines and regulations.

### 4.2. RNA Purification

Tumor RNA was isolated from 3 to 5 10 µm sections of an FFPE tissue block, applying macrodissection on unstained slides based on histopathological evaluation of a hematoxylin and eosin-stained slide to select regions with high tumor cellularity. For the isolation of normal RNA, whole sections from non-tumor tissue were used. Within two days after sectioning, the tissue sections were scraped into microcentrifuge tubes, centrifuged for 5 min at 20,000× *g*, and deparaffinized in 320 µL Deparaffinization Solution (Qiagen, Hilden, Germany) for 3 min at 56 °C on a thermomixer (500 rpm). Samples were then cooled to room temperature for 15 min. Subsequently, RNA was isolated using the miRNeasy FFPE Kit (Qiagen). Briefly, 240 µL Buffer PKD was added, and samples were mixed by vortexing. Upon centrifugation for 1 min at 11,000× *g*, 10 µL proteinase K was pipetted to the lower, clear phase. Samples were incubated for 15 min at 56 °C and 15 min at 80 °C on a thermomixer (500 rpm). Next, the lower, clear phase was transferred into a new microcentrifuge tube and incubated on ice for 3 min, followed by centrifugation for 15 min at 20,000× *g*. The supernatant was then transferred into a new microcentrifuge tube, and 25 µL DNase Booster and 10 µL DNase I stock solution was added. Samples were mixed by inverting the tubes, briefly centrifuged, and incubated for 15 min at room temperature. Subsequently, 500 µL Buffer RBC was added to the lysate, immediately followed by mixing the lysate and the addition of 1750 µL ethanol (100%). Mixed samples were loaded (per 700 µL) onto an RNeasy MinElute spin column and centrifuged for 15 s at 10,000× *g*. Flowthroughs were discarded. Upon loading, the spin column was washed twice with Buffer RPE (500 µL Buffer RPE, centrifugation for 15 s and 2 min at 10,000× *g*). Next, the spin column was dried with an open lid by centrifugation for 5 min at full speed. Finally, the FFPE tumor RNA was eluted by adding 24 µL RNase-free water directly to the spin column membrane. RNA concentrations were measured using a DropSense 96 (DropPlate-D+) fluorometer (cohort I; Unchained Labs, Pleasanton, CA, USA) or Nanodrop (cohort II; Thermo Fisher Scientific, Waltham, MA, USA). RNA integrity was assessed on a Fragment Analyzer (Agilent Technologies, Santa Clara, CA, USA) by calculating the DV200 quality metric, i.e., the percentage of RNA fragments greater than 200 nucleotides in size (Appendix A) [59].

### 4.3. mRNA Capture Sequencing

Cohort I was profiled using the TruSight RNA Pan-Cancer Panel (Illumina) using 100 ng RNA. Library preparation was performed as described in the TruSight RNA Pan-Cancer Reference Guide of Illumina. Upon the first PCR amplification, the size and purity of each library were checked on a 2100 Bioanalyzer (High Sensitivity DNA Kit; Agilent Technologies), using 1 µL library. Library concentrations were determined using Bioanalyzer software for smear analysis in the 160 bp to 700 bp range. Upon the second PCR amplification, the quality of the enriched libraries was confirmed. Libraries were quantified using qPCR according to the Illumina Sequencing Library qPCR Quantification Guide, pooled to a final concentration of 4 nM, and sequenced on a MiSeq instrument (8 libraries per flow cell; MiSeq Reagent Kit v3 (25 million 2 × 75 bp read pairs; Illumina).

Cohort II was profiled using TruSeq RNA Exome sequencing (Illumina), starting from 100 ng of RNA. Library preparation was performed as described in the TruSeq RNA Exome Guide of Illumina, with the following adaptations: fragmentation of RNA for 2 min at 94 °C, second strand cDNA synthesis for 30 min at 16 °C, and second PCR amplification using 14 PCR cycles. Upon the first and second PCR amplification, libraries were validated on a Fragment Analyzer (Agilent Technologies), using 1 µL of diluted (1:20) library. Library concentrations were determined using Fragment Analyzer software for smear analysis in the 160 bp to 700 bp range. Library quantification was qPCR-based, using the KAPA Library Quantification Kit (Kapa Biosystems, Wilmington, MA, USA). Libraries were pooled to a final concentration of 2 nM and sequenced on a NextSeq 500 instrument (20 libraries per flow cell; NextSeq 500 High Output Kit v2, 400 million 2 × 75 bp read pairs; Illumina). Data have been deposited into the European Genome-phenome Archive (EGA; EGAS00001005202).

### 4.4. Fusion Transcript Identification and Selection for RT-qPCR Analysis

Quality control on the raw sequencing reads was performed using FastQC (version 0.11.3; http://www.bioinformatics.babraham.ac.uk/projects/fastqc/ (accessed on 31 August 2019)). Trimmomatic (version 0.36; [60]) was used for adapter and 3′ end quality trimming. Subsequently, fusion transcript identification was performed using FusionCatcher (version 0.99.7c beta; hg38) [19]. FusionCatcher was used with default parameters, except for the --skip-blat option to skip BLAT aligner. First, some pre-processing and filtering steps are performed by FusionCatcher, by removing reads aligning to ribosomal RNA, among others, trimming the reads that contain adapters and poly A/C/G/T tails, and quality clipping. FusionCatcher uses multiple alignment tools (Bowtie, STAR, and Bowtie2) to increase the accuracy of alignment and fusion breakpoint detection [19]. Mapping and fusion statistics of cohorts I and II can be found in Appendix A, respectively. The number of uniquely mapped reads per sample ranges from 2,144,348 to 3,344,823 reads in cohort I and from 17,582,558 to 25,928,668 reads in cohort II.

For patients of cohort II with no pathognomonic fusion transcript detected, the list of fusion transcripts was filtered and prioritized for RT-qPCR according to the following criteria: (1) the fusion transcripts are not detected in patients with a pathognomonic fusion transcript, (2) at least one of the fusion partners is described to be part of a fusion gene or is described in the sarcoma literature, (3) likely false-positive fusion transcripts are excluded. These potential false-positive fusion transcripts have a high number of reads simultaneously mapping to both fusion partners (common mapping reads), and/or are banned by FusionCatcher, and/or have fusion partners less than 100 kbp apart.

For the identification of recurrently detected fusions in ARMS and URCS patients of cohort II, the fusions transcript lists were first filtered on the gene level, followed by exclusion of likely false-positive fusion transcripts (see exclusion criteria above), and the selection of recurrently detected transcripts with fusion events at exon-exon borders for RT-qPCR validation.

When multiple fusion transcripts of the same fusion gene partners are reported by FusionCatcher, the fusion transcript with the highest number of uniquely mapping reads encompassing the fusion junction (spanning unique reads in Appendix A) is selected for RT-qPCR validation.

### 4.5. RT-qPCR Validation of Fusion Transcripts

RT-qPCR assays for the selected fusion transcripts were designed using primerXL [61] or the Shiny app DNA Melting Thermodynamic Model, with the following salt conditions: 50 mM Na^+^, 3 mM Mg^2+^, and 1.2 mM dNTPs (http://masckareem.shinyapps.io/sbtmodel (accessed on 23 September 2019), Appendix A). To enlarge the design space for primerXL assay design, fusion transcript sequences, as reported by FusionCatcher, were extended using the flanking 5′ and 3′ exon sequences (Ensembl release 93). For RT-qPCR assay design using the Shiny app, assays were designed in the fusion transcript sequence reported by FusionCatcher. In addition, 3 reference qPCR assays (*Alu-Sq*, *Alu-Sx*, and *Alu-J*; Integrated DNA Technologies, Coralville, IA, USA) were profiled (Appendix A) [62,63]. cDNA synthesis and qPCR were performed as described in Zeka et al. [64]. Briefly, the iScript Advanced cDNA Synthesis Kit (Bio-Rad, Hercules, CA, USA) was used to prepare whole-transcriptome RT reactions from 600 ng FFPE RNA. RT products were diluted 15-fold by adding 280 µL 1× tRNA carrier (5 ng/µL, Roche, Basel, Switzerland) to 20 µL cDNA. qPCR reactions were performed in 5 µL, containing 2.5 µL SsoAdvanced Universal SYBR Green Supermix (Bio-Rad), 0.25 µL forward and reverse primer (5 µM each), and 2 µL diluted RT product. qPCR runs were performed on the LightCycler480 instrument (Roche), and data were analyzed using qbase+ version 3.2 (Biogazelle, Zwijnaarde, Belgium) [65]. mRNA capture sequencing results of each of the tumor samples were confirmed by RT-qPCR if only this sample was amplified (in at least one of the two qPCR replicates) and if all control samples (normal sample of P15 of cohort I, no template control, and carrier RNA sample) did not show amplification (based on Cq and Tm evaluation).

## Figures and Tables

**Figure 1 ijms-23-11007-f001:**
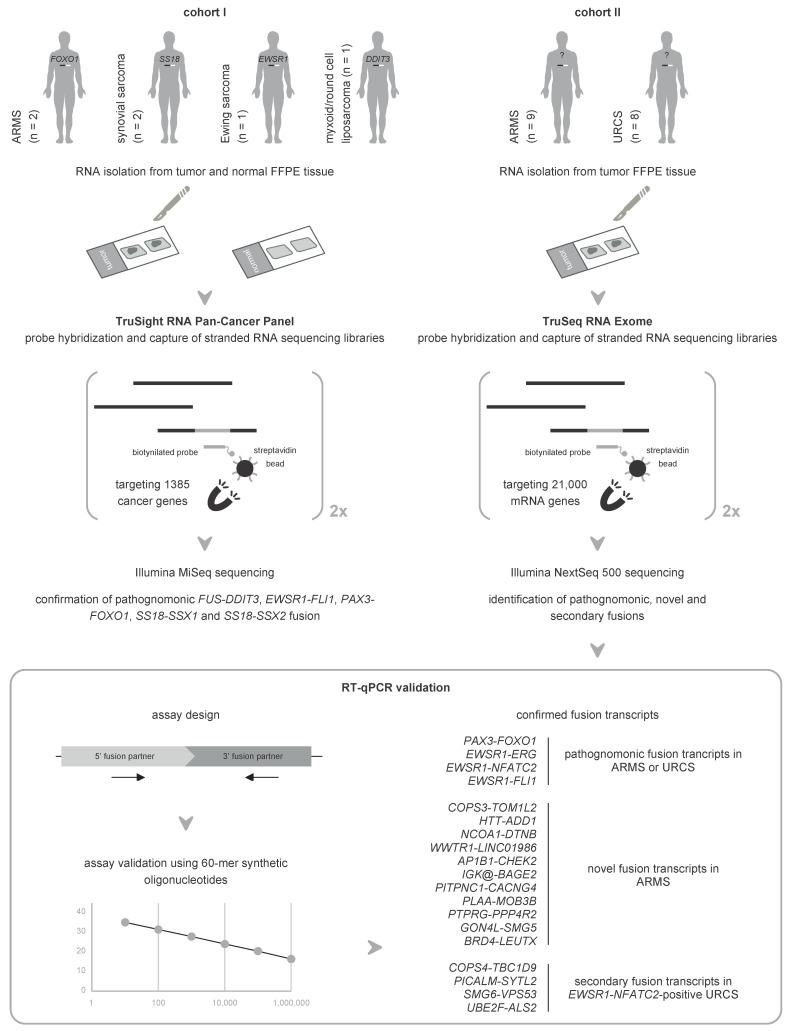
mRNA capture sequencing and RT-qPCR for the detection of pathognomonic, novel, and secondary fusion transcripts in FFPE sarcoma tissue. Cohort I, comprising six patients with a known pathognomonic fusion, is profiled using the TruSight RNA Pan-Cancer Panel. These data validated the mRNA capture sequencing analysis workflow for the identification of fusion transcripts. Subsequently, a second cohort of sarcomas that were designated fusion gene-negative by FISH analysis was analyzed using TruSeq RNA Exome sequencing. Multiple pathognomonic, novel, and secondary fusion transcripts were picked up and confirmed by RT-qPCR.

**Table 1 ijms-23-11007-t001:** mRNA capture sequencing enhances the detection of pathognomonic fusions in sarcoma. For each patient in cohort I (patient ID P11-P16) and II (patient ID P18-P29) with a pathognomonic fusion, the patient’s diagnosis is indicated, as well as the detected rearrangement by fluorescence in situ hybridization (FISH, only for patients of cohort I), and the TruSight RNA Pan-Cancer Panel (cohort I) or TruSeq RNA Exome sequencing (cohort II) results. Two different pathognomonic fusion transcripts were detected for patient P12 (same fusion transcript partners, but different chromosomal position of the 5’ end of the fusion junction) and P16 (different 3′ fusion transcript partner).

Patient ID	Diagnosis	FISH	mRNA Capture Sequencing						
			Total Number of Uniquely Mapped Reads	Fusion Transcript (5′ Partner-3′ Partner)	Spanning Pairs ^a^	Spanning Unique Reads ^b^	Read Evidence Level ^c^	Fusion Point 5′ Partner	Fusion Point 3′ Partner
P11 (cohort I)	myxoid/round cell liposarcoma	*DDIT3*	2,521,243	*FUS*-*DDIT3*	4	6	3.97	16:31,184,396:+ (end of exon)	12:57,517,753:− (start of exon)
P12 (cohort I)	Ewing sarcoma	*EWSR1*	2,837,511	*EWSR1*-*FLI1*	8	4	4.23	22:29,288,786:+ (end of exon)	11:128,807,180:+ (start of exon)
				*EWSR1*-*FLI1*	8	2	3.52	22:29,291,599:+ (end of exon)	11:128,807,180:+ (start of exon)
P13 (cohort I)	ARMS	*FOXO1*	2,144,348	*PAX3*-*FOXO1*	12	13	11.66	2:222,220,140:− (end of exon)	13:40,560,860:− (start of exon)
P14 (cohort I)	ARMS	*FOXO1*	3,344,823	*PAX3*-*FOXO1*	3	18	6.28	2:222,220,140:− (end of exon)	13:40,560,860:− (start of exon)
P15 (cohort I)	synovial sarcoma	*SS18*	2,998,047	*SS18*-*SSX1*	66	26	30.69	18:26,032,399:− (end of exon)	X:48,263,782:+ (start of exon)
P16 (cohort I)	synovial sarcoma	*SS18*	2,480,668	*SS18*-*SSX2*	11	20	12.50	18:26,032,399:− (end of exon)	X:52,700,578:− (start of exon)
				*SS18*-*SSX2B*	11	20	12.50	18:26,032,399:− (end of exon)	X:52,757,854:+ (start of exon)
P18 (cohort II)	ARMS (solid variant)	none	17,582,558	*PAX3*-*FOXO1*	11	16	1.54	2:222,220,140:− (end of exon)	13:40,560,860:− (start of exon)
P25 (cohort II)	ARMS (solid variant)	none	18,832,810	*PAX3*-*FOXO1*	13	13	1.38	2:222,220,140:− (end of exon)	13:40,560,860:− (start of exon)
P26 (cohort II)	Ewing sarcoma	none	22,414,225	*EWSR1*-*ERG*	3	5	0.36	22:29,287,134:+ (end of exon)	21:38,392,444:− (start of exon)
P27 (cohort II)	URCS; small cell osteosarcoma	none	21,846,511	*EWSR1*-*NFATC2*	19	12	1.42	22:29,282,557:+ (end of exon)	20:51,516,955:− (start of exon)
P28 (cohort II)	URCS; small cell osteosarcoma	none	18,512,302	*EWSR1*-*NFATC2*	19	13	1.73	22:29,282,557:+ (end of exon)	20:51,516,955:− (start of exon)
P29 (cohort II)	Ewing sarcoma	none	21,344,451	*EWSR1*-*FLI1*	4	5	0.42	22:29,287,134:+ (end of exon)	11:128,805,366:+ (start of exon)

^a^ Number of paired-end reads spanning the fusion, but not directly encompassing the fusion junction (including multi-mapping reads). ^b^ Number of uniquely mapping reads encompassing the fusion junction; also known as split reads. ^c^ Number of fusion supporting reads per million uniquely mapped reads.

**Table 2 ijms-23-11007-t002:** mRNA capture sequencing identifies novel fusion transcripts in sarcoma. For patients in cohort II (unique patient ID), the TruSeq RNA Exome sequencing results of the prioritized novel fusion transcripts for RT-qPCR validation are listed. RT-qPCR confirmed transcripts are indicated in gray (light gray for assays designed using primerXL; dark gray for assays designed using the Shiny app DNA Melting Thermodynamic Model).

Patient ID	Diagnosis	mRNA Capture Sequencing						
		Total Number of Uniquely Mapped Reads	Fusion Transcript (5′ Partner-3′ Partner)	Spanning Pairs ^a^	Spanning Unique Reads ^b^	Read Evidence Level ^c^	Fusion Point 5’ Partner	Fusion Point 3’ Partner
P17	ARMS	17,891,296	*COPS3*-*TOM1L2*	11	9	1.12	17:17,276,035:− (end of exon)	17:17,898,674:− (start of exon)
			*HTT*-*ADD1*	7	5	0.67	4:3,174,799:+ (end of exon)	4:2,904,764:+ (start of exon)
			*NCOA1*-*DTNB*	6	9	0.84	2:24,728,476:+ (end of exon)	2:25,531,597:− (start of exon)
P19	ARMS	20,614,167	*AC138409.2*-*NAIP*	3	4	0.34	5:34,175,822:− (end of exon)	5:71,003,857:− (start of exon)
			*NLRP2*-*RPL36A*	3	4	0.34	19:54,973,997:+ (in exon)	X:101,395,351:+ (in exon)
			*RPL36A*-*NLRP2*	3	5	0.39	X:101,391,808:+ (in exon)	19:54,973,973:+ (in intron)
			*WWTR1*-*LINC01986*	6	5	0.53	3:149,572,864:− (end of exon)	3:23,968:+ (start of exon)
P20	ARMS	18,472,958	*AP1B1*-*CHEK2*	3	2	0.27	22:29,349,218:− (end of exon)	22:28,710,059:− (start of exon)
			*IGK@*-*BAGE2*	54	17	3.84	2:89,631,593:− (end of exon)	21:10,499,475:+ (start of exon)
			*PITPNC1*-*CACNG4*	3	2	0.27	17:67,532,950:+ (end of exon)	17:67,018,189:+ (start of exon)
			*PLAA*-*MOB3B*	8	3	0.60	9:26,910,338:− (end of exon)	9:27,330,616:− (start of exon)
			*PTPRG*-*PPP4R2*	3	3	0.32	3:61,562,372:+ (end of exon)	3:72,998,077:+ (start of exon)
			*WDR74*-*ACTB*	3	3	0.32	11:62,837,505:− (in intron)	7:5,527,639:− (in exon)
P22	ARMS	17,675,699	*GON4L*-*SMG5*	2	5	0.40	1:155,804,949:- (end of exon)	1:156,253,508:- (start of exon)
P23	ARMS	18,512,177	*BRD4*-*LEUTX*	3	11	0.76	19:15,254,152:− (end of exon)	19:39,784,527:+ (start of exon)
P24	ARMS	19,914,559	*ATN1*-*MAML2*	3	9	0.60	12:6,936,716:+ (in exon)	11:96,092,266:− (in exon)
P30	URCS	23,211,375	*NCOA3*-*TBP*	3	3	0.26	20:47,651,098:+ (in exon)	6:170,561,926:+ (in exon)
			*TBP*-*NCOA3*	3	8	0.47	6:170,561,939:+ (in exon)	20:47,651,071:+ (in exon)
P32	URCS	20,217,802	*MEST*-*RGS22*	2	2	0.20	7:130,500,532:+ (end of exon)	8:100,093,509:− (start of exon)
P33	URCS	25,928,668	*AC009021.1*-*RRN3*	2	2	0.15	16:22,613,454:− (end of exon)	16:15,080,096:− (start of exon)

^a^ Number of paired-end reads spanning the fusion but not directly encompassing the fusion junction (including multi-mapping reads). ^b^ Number of uniquely mapping reads encompassing the fusion junction; also known as split reads. ^c^ Number of fusion supporting reads per million uniquely mapped reads. ARMS: alveolar rhabdomyosarcoma; URCS: undifferentiated round cell sarcoma.

**Table 3 ijms-23-11007-t003:** mRNA capture sequencing detects recurrent fusion transcripts in sarcoma. For patients in cohort II (unique patient ID), the TruSeq RNA Exome sequencing results of the recurrently detected novel fusion transcripts are listed, excluding falsely identified fusion transcripts and selecting transcripts with fusion events at exon-exon borders (see Section 4). RT-qPCR confirmed transcripts are indicated in gray (assays designed using the Shiny app DNA Melting Thermodynamic Model).

Recurrently Detected Fusion (5′ Partner-3′Partner)	Disease	Patient ID	mRNA Capture Sequencing					
			Total Number of Uniquely Mapped Reads	Spanning Pairs ^a^	Spanning Unique Reads ^b^	Read Evidence Level ^c^	Fusion Point 5’ Partner	Fusion Point 3’ Partner
*AC245595.1*-*IGK@*	ARMS	P19	27,247,781	10	2	0.44	1:144,250,225:− (end of exon)	2:89,581,140:− (start of exon)
		P25	25,800,361	59	2	2.36		
*ELMO1*-*AOAH*	ARMS	P18	23,669,117	2	3	0.21	7:36,861,659:− (end of exon)	7:36,674,009:− (start of exon)
		P25	25,800,361	3	6	0.35		
*COPS4-TBC1D9*	URCS	P27	28,739,150	4	3	0.24	4:83,035,298:+ (end of exon)	4:140,679,843:− (start of exon)
		P28	24,863,986	4	4	0.32		
*PICALM-SYTL2*	URCS	P27	28,739,150	3	2	0.17	11:85,974,708:− (end of exon)	11:85,758,114:− (start of exon)
		P28	24,863,986	2	3	0.20		
*SMG6-VPS53*	URCS	P27	28,739,150	121	23	5.01	17:2,172,658:− (end of exon)	17:710,613:− (start of exon)
		P28	24,863,986	44	21	2.61		
*UBE2F-ALS2*	URCS	P27	28,739,150	6	13	0.66	2:237,973,225:+ (end of exon)	2:201,718,210:− (start of exon)
		P28	24,863,986	5	8	0.52		

^a^ Number of paired-end reads spanning the fusion but not directly encompassing the fusion junction (including multi-mapping reads). ^b^ Number of uniquely mapping reads encompassing the fusion junction; also known as split reads. ^c^ Number of fusion supporting reads per million uniquely mapped reads. ARMS: alveolar rhabdomyosarcoma; URCS: undifferentiated round cell sarcoma.

## Data Availability

RT-qPCR data generated during this study are included in this published article and its supplementary information files. mRNA capture sequencing data are available in the European Genome-phenome Archive repository (EGA; EGAS00001005202).

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
