# Peer review of "mRNA Capture Sequencing and RT-qPCR for the Detection of Pathognomonic, Novel, and Secondary Fusion Transcripts in FFPE Tissue: A Sarcoma Showcase"

_ijms, 2022, doi:10.3390/ijms231911007_

Round 1
Reviewer 1 Report
Within the manuscript “mRNA capture sequencing and RT-qPCR for the detection of pathognomonic, novel, and secondary fusion transcripts in FFPE tissue: a sarcoma showcase”, the authors present a detailed evaluation of fusion gene detection based on Illumina RNA sequencing methods and validation by RT-qPCR.
Though a small cohort of a subset of sarcoma patients, the study represents a neat demonstration of identifying pathognomonic and secondary fusion events from targeted RNA sequencing datasets. The study also highlights the advantages of using this technique above classical fusion detection methods, such as FISH.
However, there are a few key issues relating to the manuscript that require elaboration or further comment for clarity –
Results
Orthogonal validation of known fusions in sarcoma using mRNA capture sequencing: Was an attempt made to repeat the normal tissue library prep for patient P16 or was it just discarded at the first failure to achieve target region enrichment? Could the authors comment on why they believe this sample failed?
What additional detection threshold criteria were applied for fusion gene exclusion within the FusionCatcher results list? For example, was it a specific / minimum spanning pair number, spanning unique read number, or specific read evidence level? Or were all fusions considered plausible, provided they passed the criteria laid out in “4: Materials and Methods – Fusion transcript identification and selection for RT-qPCR analysis”?
Unbiased mRNA capture sequencing reveals pathognomonic fusion genes in clinicopathological enigmatic sarcomas: If FISH initially demonstrated 0% positive tumour cells, could the authors detail their rationale as to the detection of fusion events in the ARMS and URCS patients? For example, was the fusion position identified by sequencing incompatible with the FISH probes for FOXO1, FUS, CIC, and BCOR?
Novel fusion transcripts are identified and validated in clinicopathological enigmatic sarcomas: Could the authors comment on the positive RT-qPCR results in the control samples, and how they interpret this observation?
The majority of the RT-qPCR-confirmed fusion transcripts are the result of intrachromosomal rearrangement. Could the authors elaborate further on this observation and why this particular event is so prevalent in their results.
RT-qPCR confirms recurrently detected secondary fusion transcripts in sarcomas with an EWSR1-NFATC2 fusion: It’s intriguing that several identical secondary fusion transcripts were identified within patients P27 and P28. Moreso, as they also share the same pathognomonic fusion gene. Given the unique mechanism that would be required to precisely generate all of these fusion genes identically in both patients, did the authors undertake any additional sequencing analyses to rule out the possibility of sample contamination specifically between these two patients?
Discussion
In the Introduction, the authors state that the prediction accuracy of different fusion-calling bioinformatic software varies considerably, which is reflected in the variable detection of false positive fusions. Could the authors comment on their specific use of FusionCatcher alone for their data analysis, rather than utilising multiple fusion-calling software in parallel to assist in refining the final fusion gene list?
The authors highly recommend prioritising exon-exon fusions for RT-qPCR validation. Yet later in the text, they refer to identifying lncRNA fusions (Patient P19), which would be excluded by this style of analysis. Could the authors comment on this apparent contradiction?
Materials and Methods
mRNA capture sequencing: The authors nicely detailed the sequencing platforms utilised for this manuscript and the level of multiplexing undertaken per sequencing run. Whilst included in Supplementary Table 1, could the authors also provide the range of unique mapped reads achieved within the methods section to improve clarity?
Tables
Table 1: A better method of distinction should be applied to clearly identify patients from Cohort 1 and Cohort 2 (similar to the style used in Table 3).
Supplementary Table 1: There’s a typo of “Fustion” instead of “Fusion” for both spreadsheets.
Figures
Supplementary Figure 1: Unfortunately, I couldn’t access this figure.
Reviewer 2 Report
The manuscript “mRNA capture sequencing and RT-qPCR for the detection of 2 pathognomonic, novel, and secondary fusion transcripts in 3 FFPE tissue: a sarcoma showcase” by Decock et al. describes the use of RNA-seq capture sequencing coupled with the FusionCatcher software for validating and further exploring detection of pathognomonic fusions in FFPE tumor samples. Using a TruSeq RNA Exome capture approach, the authors identified pathogenic fusions in samples that were otherwise FISH negative, coupled with secondary fusion transcript detections potentially of interest. By doing so, the authors further demonstrate the utility of whole exome capture RNA sequencing for characterization of tumor FFPE samples, particularly when other approaches lack sensitivity for detecting important oncogenic fusions.
The authors’ method for detection of fusions based on capture sequencing exclusively relies on FusionCatcher. It would be useful for the authors to explore including additional methods such as Arriba or STAR-Fusion as part of their approach in order to further boost fusion detection sensitivity. I suspect that the experimentally validated fusions are more likely to have been predicted by multiple such methods. Also, it would be useful to know if there are cancer samples that other methods find evidence of oncogenic fusions that were lacking from FusionCatcher.
For sequencing-based validation of samples harboring known fusions, the TruSight RNA Pan-Cancer panel was leveraged, and only for those samples lacking the expected pathognomonic fusions was the full exome capture-sequencing panel used. It would be useful for the authors to comment on the advantages and disadvantages of each approach in the manuscript, and whether or not a full exome capture approach might suffice for cancer transcriptome sequencing in a clinical setting.
Reviewer 3 Report
Decock et al., reported mRNA capture sequencing for the detection of fusion transcripts using clinical FFPE samples. They were able to detect known fusions with cohort one using TruSight RNA pan-cancer panel. They then used TruSeq RNA Exome platform to detect fusion transcripts on samples with no known fusions. The results are interesting, and the study is timely. I only have a few minor concerns/suggestions.
1. Quite some reports have demonstrated that fusion transcripts can be made due to intergenic splicing. The term “gene fusion” or “chimeric genes” is confusing since it indicates events happening at DNA level. All the fusions here are uncovered by RNA-Seq with no evidence whether they are products of chromosomal rearrangement. It is more accurate to call them “chimeric RNA” or “fusion transcripts”.
- It is interesting that the authors discovered fusion PAX3-FOXO1 and EWSR ones in clinical cases with negative FISH signal. PAX3-FOXO1 was detected in situations with no t(2;13) and was believed to be product of trans-splicing (PMID: 24089019). This work suggests the potential of situation that maybe over active trans-splicing may be an alternative mechanism to produce the fusion transcript. There is also a hypothesis that fusion RNA may come first to mediate chromosomal rearrangement. This possibility should be discussed.
- Have the authors done the same fusion detection with normal margins for cohort 2? What are the results? Can the same fusions be detected there?
Round 2
Reviewer 1 Report
All points raised have been addressed.
Reviewer 2 Report
To address the issue of not running other fusion prediction methods than FusionCatcher, the authors counterproductively added the text "g. To reduce the risk on reporting results of false positive novel fusion transcripts, results of multiple fusion callers are often combined to pinpoint a common set of transcripts. However, this still does not exclude the possibility of detecting false positives and might even counteract the performance of highly accurate fusion callers by excluding true positives not identified by less accurate fusion callers. For example, the experimentally validated fusion CSE1L-AL035685.1 in SK-BR-3 cells is only predicted by FusionCatcher and not by 22 other fusion callers [17]. Therefore, in this study, we opted to make use of single fusion caller (i.e. FusionCatcher, that previously showed to be fast and highly accurate [17,19–21]), combined with a literature-based priorization strategy to select fusion transcripts for RT-qPCR confirmation in order to truly distinguish true and false positive fusions."
While FusionCatcher is found to be an excellent method and can capture fusions not found by others, it is not devoid of false negatives that are captured by other prediction methods. In a clinical sequencing strategy, it would be warranted to apply multiple methods and not restrict yourself to only the overlapping predictions, but rather to use all predictions (the union as opposed to the intersection) for defining candidates worth pursuing for experimental validation. Those that are predicted by multiple methods are more likely to validate, but having multiple overlapping predictions need not be a defining characteristics for targeting fusions for validation. All other clinical groups I'm aware of leverage multiple such methods in their approach for this specific reason.
By specifically limiting fusion predictions to FusionCatcher, you may be missing fusion candidates worthy of validation, some of which may be clinically relevant.
This issue should be properly addressed in the manuscript, as opposed to the authors current strategy aimed at defending their sole use of FusionCatcher as the oracle for defining the landscape of potentially clinically relevant fusions in patient samples.
If the authors should choose to not examine the results of other alternative predictors such as STAR-Fusion and Arriba (both independently validated as among the best methods - PMID:34146471), then the authors should at the very least properly address the issue of potential false negatives in their current design through revisions to the manuscript text accordingly.
Reviewer 3 Report
The authors have addressed all my concerns. Recommend acceptance.
